# Elevation Angle Estimations of Wide-Beam Acoustic Sonar Measurements for Autonomous Underwater Karst Exploration

**DOI:** 10.3390/s20144028

**Published:** 2020-07-20

**Authors:** Yohan Breux, Lionel Lapierre

**Affiliations:** Laboratory of Informatics, Robotics and MicroElectronics (LIRMM) (UMR 5506 CNRS—UM), Université Montpellier, 161 rue Ada, CEDEX 5, 34392 Montpellier, France; lapierre@lirmm.fr

**Keywords:** karst, underwater, sonar, SLAM, AUV, 3D

## Abstract

This paper proposes a solution for merging the measurements from two perpendicular profiling sonars with different beam-widths, in the context of underwater karst (cave) exploration and mapping. This work is a key step towards the development of a full 6D pose SLAM framework adapted to karst aquifer, where potential water turbidity disqualifies vision-based methods, hence relying on acoustic sonar measurements. Those environments have complex geometries which require 3D sensing. Wide-beam sonars are mandatory to cover previously seen surfaces but do not provide 3D measurements as the elevation angles are unknown. The approach proposed in this paper leverages the narrow-beam sonar measurements to estimate local karst surface with Gaussian process regression. The estimated surface is then further exploited to infer scaled-beta distributions of elevation angles from a wide-beam sonar. The pertinence of the method was validated through experiments on simulated environments. As a result, this approach allows one to benefit from the high coverage provided by wide-beam sonars without the drawback of loosing 3D information.

## 1. Introduction

The context of this paper is the development of robotic systems to explore karstic environments. Karsts are networks of underground natural conduits resulting from the dissolution of soluble rocks, limestone, dolomite and gypsum that drain groundwater on large scales. This is illustrated in Figure 1. Karst aquifers are of upmost importance as they are the main sources of drinking water for millions of people worldwide. They are gigantic tanks of renewable fresh water, but their exploitation requires deeper investigation of their characteristics: location, geomorphology and dynamics. Active groundwater management is a major issue for public authorities involved in the prospection, protection and management of the groundwater resource and hydrogeologic risk prevention. In order to have a better understanding of karst aquifers, it is essential to know the geometry of the flow paths, which may be difficult to acquire. Currently, this is done by cave divers with a limited range due to physiological constraints. An autonomous robotic solution would provide the capacity to go further and deeper in the karst conduit while acquiring dense information on the environment.

However, there are still many scientific and technical challenges to overcome before reaching this goal. This paper is related to the robot localization problem in a SLAM framework. Recently, a lot of studies have focused on vision-based localization approaches. Although they can provide accurate and dense measurements, they are subject to the water conditions and the appearance of the environment. Thus, for extensive exploration of an unknown environment, we need to complement it with robust sensors. For terrestrial robots, one can use a scan matching technique such as ICP (iterative closest point) using LIDAR [1]. However, in underwater robotics, LIDAR requires water transparency, which is not guaranteed in a karstic environment. Then, the solution relies on acoustic sonars. This has two major drawbacks:
As sonar deals with acoustic waves, the time-of-flight is always higher than for a laser. Moreover, mechanical pencil-beam sonars are slow, and unlike LIDAR, for which full scan are instantaneous, they take several seconds.A sonar beam can also have non-negligible beam width which leads to high uncertainty on the 3D positions of the observations.


Our final objective was then to develop a variant of ICP adapted to those constraints. In particular, that requires dealing with the 3D estimation of sonar measures with non-negligible beam width. In our setup, similarly to [2,3,4], the robot was mounted with two mechanically scanning sonar systems (MSIS). A narrow-beam sonar is mounted vertically while the another wide-beam sonar is mounted horizontally. In [4], the vertical sonar is only used for 3D reconstruction of the environment while the horizontal sonar is used to estimate the 2D robot motion using a probabilistic ICP approach. We introduce in this paper a new approach for merging both sonars’ information to estimate the 3D positions of points in the environment measured with the wide-beam sonar. Indeed, the advantage of a wide acoustic beam is to obtain a large coverage of the environment, increasing the surface covered by two successive full 360° scans. Obviously, it comes at the price of having a high uncertainty in the true observed 3D point. The idea was to use the narrow vertical sonar to learn a local probabilistic model of the surface with Gaussian process regression. This model was then leveraged to estimate the probabilistic distribution of each horizontal sonar measurement.

The first contribution of this paper is to propose a 3D Gaussian process regression adapted to karst environment and more generally applicable to approximately elliptic cylindrical shaped surfaces. We analyzed the results obtained with different kernels to better comprehend their effects on the estimated surfaces. The second contribution is a principled approach to estimating the scaled-beta distribution of elevation angles from a wide-beam sonar by exploiting the previously estimated environment. The pertinence of the method was quantitatively assessed through simulations wherein the estimated distributions could be evaluated against the environment ground truth.

The structure of the paper is as follows. Section 2 gives an overview on related works in underwater robotics. Our robotic system and related definitions used throughout the paper are described in Section 3. Section 4 is the main part of this paper in which we develop our approach. Finally, Section 5 describes evaluations of our work in simulated environments. The conclusion and future working directions wrap up in Section 6.

## 2. Related Works

In the last few decades, a wide range of simultaneous localization and mapping (SLAM) techniques have been developed [1,6]. In particular, a large focus has been put in the robotic field on terrestrial and aerial applications based on camera sensors [7,8]. Some efforts were also made to use optical sensors for underwater applications [9,10,11], but were rather limited as they were highly dependent on water conditions (turbidity, luminosity). Hence we cannot guarantee the robot localization in an unknown environment based solely on vision sensors. However, those approaches can be used when applicable to complement more robust sonar-based methods.

Thus, a large part of the underwater SLAM literature is based on acoustic sonars. References [12] and [13] proposed a pose-graph SLAM approach using a dual-frequency identification sonar (DIDSON) for the inspection of underwater structures. The advantage of their approach is to compensate for the low coverage of their sonar by sweeping it in a stationary position. However, this is not applicable in the context of karst exploration where the robot moves while sensing the environment. Closer to our application, Fairfield et al. [14,15] first developed a SLAM approach for cenote (underwater vertical tunnel) exploration. It is based on a Rao-Blackwellized particule filter (RBPF) with a mapping is represented by a 3D evidence grid in an octree structure. They used an array of 54 pencil-beam sonars placed around their vehicle in order to map the environment and recognize previously mapped region for SLAM purposes. Rahman et al. [10] proposed a system for underwater cave exploration. They notably leverage profiling sonar, and inertial and depth sensors to strengthen their approach based on stereo camera vision. However, the system is not designed to work in degraded mode when vision is not available.

Palomer et al. [3] used an extended Kalman filter (EKF) SLAM framework for seafloor mapping. They used a multi-beam echosounder and the resulting scans were matched using probabilistic ICP [16], as proposed by [2] in a similar context. While this is pertinent for bathymetry applications, we cannot fix a multibeam echosounder in a fixed direction for the exploration of a karst aquifer. Similarly, Burguera [4] developed a method for registering data from mechanically scanned imaging sonar (MSIS). Elevation angles from beam measurements and thus 3D data are obtained through the assumption that the robot moves parallel to the seafloor. Obviously we cannot make the same assumption in our context. Chen et al. [17] also recently proposed a Rao-Blackwellized particle filter (RBPF) SLAM framework using MSIS for the same application. They considered a beam enpoint model to compute the likelihood of an observation which is classical in the occupancy grid approach. They focus on the 2D localization of the robot and the lost of 3D information from the wide-beam sonar was ignored.

Unlike previous works which focused either on inspection or bathymetry in open space, we are interested in the exploration and mapping of confined underwater environments (karstic caves). Mallios et al. [18,19] presented experimental results of the exploration of an underwater cave based on their earlier work [2]. They also provided a dataset from this experiment. Thanks to the MSIS vertical beam width and by supposing a quasi-translational displacement between two full 360° scans, they could restrict the estimation to a 2D problem. Depth, pitch and roll absolute values were directly read from the precision and the IMU sensor. Our work was inspired by this approach and aimed at considering the full 3D problem. For instance, this approach cannot deal with L-shaped conduits where measured points cannot be consider coplanar. Furthermore, even in coplanar cases, it is not theoretically sound to separate the rigid transformation estimation to SE(2) and then set the other values (depth, pitch and roll) with the sensor’s data. The problem should be cast as a constrained optimization on SE(3). Finally, Martins et al. [20] developed a robotic system called UX-1 under the European project UNEXMIN designed to explore flooded, abandoned mines. It is equipped with a camera, a four-customized-laser-based structured light system and multibeam profiler sonar. They extract natural landmarks from the different sensors which are then integrated into a graph-based SLAM framework. While their context was similar to ours, they mainly relied on the fusion of vision and a structured light system. Note that while we focus solely on sonar sensors in this paper, we ultimately aim at complementing already existing methods.

To the best of our knowledge, karstic exploration by AUV remains rather unexplored territory. The main difficulty is that the karst can be composed with narrow galleries or large caves. In particular, we need MSIS sonar with a wide beam to increase the chance that a previously mapped region will be encountered when revisited. Meanwhile, we also need 3D data from the sonar as we cannot restrict the problem to 2D here. Indeed, in a karstic environment, the robot would have to follow non-planar trajectories. Furthermore, it gives the possibility to further improve the relative displacement estimation by considering a 3D ICP constrained in depth, pitch and roll. This means that we have to estimate the elevation angle from the sonar measurements. It is the same problem expressed recently by Wang et al. [21]. Their work was similar to [3] but improved on it by searching to estimate the sonar elevation angles which were assumed null in the previous works. They did so by directly tracking features in the multibeam sonar images. They also estimated the terrain height by modeling it with Gaussian processes directly integrated in their factor graph SLAM. Note that in their case, the elevation angle was indirectly obtained through the terrain height estimate. In our case, we used MSIS (single beam) sonar. Besides, due to the karstic environment, we had to fully model the environment. The assumptions, such as elevation angle with null mean, are not applicable in our case.

## 3. Definitions

### 3.1. Notation

We define here the notation used throughout this paper. Vectors are represented in bold lower case (e.g., x) and matrices in upper case (e.g., M). Integer intervals are denoted 〚a,b〛. If *X* is a random variable, X∼q(λ) means that *X* follows a law defined by the probability density function (pdf) *q* with parameters λ. With an abuse of notation, we denote as P(X=x)=P(x) the probability of *X* taking the value *x* (in the continuous case , P(X=x) is always zero (null set). In this case, $P\left(X=x\right)\;\hat{=}\;q\left(x|\lambda \right)$ where *q* is its pdf.). We also denote $\hat{X}$ the estimator of a random variable *X*.

The operator ⊕ (resp. ⊖) represents the pose-pose and pose-point composition (resp. inverse composition) operator [22]. We use the notation $x_{\left|\{R\right\}}$ to express that a vector x is expressed in the basis (frame) {R}.

Let *M* be a *d*-manifold embedded in $R^{n}$. We denote $T_{x}M$ the tangent plane (*d*-dimensional vector space) at x∈M. In particular, if *M* is a Riemaniann manifold, we denote $<.,.{>}_{x}$ the inner product on $T_{x}M$.

SE(3) is the special Euclidean group representing the rigid transformations of space. It is a Lie group and we denote se(3) its Lie algebra. We denote Euler angles ${\left[\psi ,\chi ,\phi \right]}^{T}$ in yaw–pitch–roll convention. R(q) (resp. R(ψ,χ,ϕ)) is the rotation matrix corresponding to a quaternion q (resp. the Euler angles ${\left[\psi ,\chi ,\phi \right]}^{T}$). When needed, we also use the Matlab notation x(i:j).

### 3.2. System Description

In the sequel we will consider the underwater robotic system Telemaque, depicted in Figure 2a. Telemaque is equipped with 8 thrusters, affording the robot with omni-directional capability. A complete sensor suite (9-axis IMU and DVL) allows for computing dead-reckoning navigation. Two profiling sonars horizontally and vertically scan the environment and will be used for SLAM. As a preliminary step, the merging between the 2 sonar measurements is performed. This is the subject of this paper.

According to Figure 2b, we define several frames:
Let {0} be the universal fixed frame;Let {B} be the body-fixed frame, attached to the robot;Let $\left\{S^{H}\right\}$ be the frame attached to the horizontal profiling sonar;Let $\left\{S^{V}\right\}$ be the frame attached to the vertical profiling sonar.


Equipped with those frames definitions, let:
τ be the (3×1) vector denoting the position of {B} origin with respect to {0};q be the (4×1) normalized quaternion denoting the attitude of {B} with respect to {0};$\eta ={\left[\tau^{T}\;q^{T}\right]}^{T}$ be the (7×1) vector denoting the system pose with respect to {0};$p^{H}={\left[\tau_{H}^{T}\;q_{H}^{T}\right]}^{T}$ be the (7×1) vector denoting the pose of the horizontal profiling sonar, with respect to {B}, i.e., position and attitude of $\left\{S^{H}\right\}$ with respect to {B};$p^{V}={\left[\tau_{V}^{T}\;q_{V}^{T}\right]}^{T}$ be the (7×1) vector denoting the pose of the vertical profiling sonar, with respect to {B}, i.e., position and attitude of $\left\{S^{V}\right\}$ with respect to {B}.


The robot is equipped with navigation sensors (IMU and DVL) which allow for performing dead-reckoning navigation and provide, regularly, an estimation of the global system pose η. It is assumed to follow a normal Gaussian distribution, i.e., $\eta ∼N(\bar{\eta },\Sigma_{\eta })$, where $N(\bar{\eta },\Sigma_{\eta })$ stands for a Gaussian distribution of mean $\bar{\eta }$ and covariance matrix $\Sigma_{\eta }$. We call discretized trajectory the set of system pose $H=\left\{\eta_{i}\right\},i\in 〚1,n^{r}〛$ regularly obtained by dead-reckoning along its $n^{r}$ first poses.

### 3.3. Sonars and Measurements

The two sonars embed an acoustic rotating head and are mounted on the robot according to Figure 3 with poses in {B} denoted $p^{H}$ and $p^{V}$. We denote $\Theta^{H}$ (resp. $\Theta^{V}$) the vertical aperture and $\Psi^{H}$ (resp. $\Psi^{V}$) the horizontal aperture for the horizontal (resp. vertical) sonar.

Due to those apertures, a single beam from those sonars can correspond to several echos with different ranges, as illustrated in Figure 4. In our case, according to the specification of the sonar used, we consider $\Psi^{H}=\Psi^{V}=\Theta^{V}=0$. In other words, only the vertical aperture of the horizontal sonar is taken in account. As a corollary, a single beam from the vertical sonar provides a unique measure $m_{i}^{v}=\left[\rho_{i}^{v}\;0\;\psi_{i}^{v}\right]$ expressed as spherical coordinates in its local frame $\left\{S^{V}\right\}$ and acquired from the robot pose $\eta_{i}$ (in fact, it is slightly different from the classic spherical coordinates in the θ definition. Here it is defined as the oriented angle between the XY-plane and the measured ray. See Figure 3.

For the horizontal sonar, due to the vertical aperture, a single insonification can provide *n* different echos corresponding to *n* different ranges $\rho_{j}$ with unknown elevation angles $\theta_{j},\;j\in 〚1,n〛$. Hence the *i*-th measurement of the horizontal sonar is represented by a 3×n matrix $M_{i}^{h}=\left[m_{ij}^{h}\right]$ where each $m_{ij}^{h}={\left[\rho_{ij}^{h}\;\theta_{ij}^{h}\;\psi_{i}^{h}\right]}^{T}$ corresponds to an observed point on the environment surface. Note that the angle $\psi_{i}^{h}$ corresponding to the sonar head rotation angle is the same for the *n* points coming from the *i*-th beam.

Similarly to [4], only the horizontal sonar can be used to estimate the relative poses using a variant of ICP. Thus we define a complete scan $S=\{M^{h},M^{v}\}$ when the horizontal scan has completed a full 360° measurement. $M^{h}=\left\{M_{i}^{h}\right\},\;i\in 〚1,n^{h}〛$ is the set of measurements from the horizontal sonar during a complete scan. Similarly, we define $M^{v}=\left\{m_{k}^{v}\right\},\;k\in 〚1,n^{v}〛$ the set of measurements from the vertical sonar obtained during the same interval of time.

Note that in [4] the vertical sonar is only used to add points to the reconstructed 3D environment. In this work, the vertical sonar is also used to estimate the elevation angles from the horizontal sonar.

As the dead-reckoning navigation and both sonar measurements are asynchronous, the *i*-th sonar measurements $M_{i}^{h},M_{i}^{v}$ are not acquired simultaneously at the *i*-th robot pose $\eta_{i}$. We then define the following mapping linking the sonar measurements to the robot pose at which they have been acquired. We denote $\sigma^{h}:〚1,n^{h}〛\to 〚1,n^{r}〛$ (resp. $\sigma^{v}:〚1,n^{v}〛\to 〚1,n^{r}〛$) the function which returns the pose index at which a horizontal (resp. vertical) sonar measurement has been made. This means for example that the *i*-th horizontal measurement $M_{i}^{h}$ has been acquired at the robot pose $\eta_{\sigma^{h}\left(i\right)}$.

We also define the functions $g^{h}$ and $g^{v}$ which maps local spherical coordinates in the horizontal (resp. vertical) sonar frame to the Cartesian coordinates in the global frame {0}:
(1)$$\begin{array}{c} g^{h}:\left\{\begin{array}{cc} R^{+}\times \Theta^{H}\times \left[-\pi ,\pi \right]\times 〚1,n^{h}〛 & \to \Omega \\ \left[\begin{array}{c} r\\ \theta \\ \psi \end{array}\right],\;i & \mapsto \left[\begin{array}{c} x\\ y\\ z \end{array}\right]=\eta_{\sigma^{h}\left(i\right)}⊕p^{H}⊕\left[\begin{array}{c} r\cos(\theta )\cos(\psi )\\ r\cos(\theta )\sin(\psi )\\ r\sin(\theta ) \end{array}\right] \end{array}\right. \end{array}$$
(2)$$\begin{array}{c} g^{v}:\left\{\begin{array}{cc} R^{+}\times \Theta^{V}\times \left[-\pi ,\pi \right]\times 〚1,n^{v}〛 & \to \Omega \\ \left[\begin{array}{c} r\\ \theta \\ \psi \end{array}\right],\;i & \mapsto \left[\begin{array}{c} x\\ y\\ z \end{array}\right]=\eta_{\sigma^{v}\left(i\right)}⊕p^{V}⊕\left[\begin{array}{c} r\cos(\theta )\cos(\psi )\\ r\cos(\theta )\sin(\psi )\\ r\sin(\theta ) \end{array}\right] \end{array}\right. \end{array}$$
where Ω is the environment surface. All the definitions are illustrated in Figure 2.

### 3.4. Gaussian Process Regression for Surface Estimation

A Gaussian process (GP) [23] is a set of random variables such that every finite subset of those variables are normally distributed. A GP regression is thus a non-parametric method for function regression. In our case, it is used to model the environment surface through the relation between the vertical sonar output (range $\rho^{v}$) and inputs (rotation angle $\psi^{v}$ and curvilinear abscissa *s* along the robot’s trajectory as explained in the Section 4.1):
(3)$$\begin{array}{c} f\left(s,\psi^{v}\right)+\epsilon =\rho^{v}∼N\left(\bar{\rho },\sigma_{\rho }^{2}\right) \end{array}$$
(4)$$\begin{array}{c} f∼\mathrm{GP},\;\epsilon ∼N(0,\sigma_{n}^{2}) \end{array}$$
where ϵ is the observation noise. Let $f_{p}$ denote the prior function which models our prior knowledge of the environment. The definition of this function in our case will be discussed in Section 4.2.1.

Gaussian processes involve kernel functions which measure the similarity between data points. A kernel is simply defined as a function mapping a pair of inputs $x,x^{\prime }$ into R. In the case of Gaussian processes, the notion of similarity is defined by the covariance. Hence we are interested in symmetric positive definite kernels. More details on Gaussian processes and kernel properties can be found in [23]. The choice of the kernel has a great incidence on the training result. The radial basis function (RBF) kernel is generally the default kernel used in GP:
(5)$$K^{RBF}\left(x,x^{\prime }\right)=\sigma_{p}^{2}e^{\left(-\frac{||x-x^{\prime }{\left|\right|}^{2}}{2l^{2}}\right)}$$
where $\sigma_{p}^{2}$ is the output length scale and $l^{2}$ is the input length scale. It gives smooth results which are appropriate for smooth surface but not for rougher one. This kernel is member of the broader Matern class of covariance functions $K_{\nu }^{M}$ [24]. Those kernels depend on a positive parameter ν which controls the smoothness of the function. Only values of the form $\nu =p+\frac{1}{2},p\in N$ are considered, as it allows one to considerably simplify the general expression of Matern kernel. Besides, values above $\frac{7}{2}$ are ignored as they give results almost identical to the case ν=+∞ (see [23], pp. 84–85). In particular, the case $\nu =\frac{1}{2}$ corresponds to the squared exponential kernel
$$K^{exp}\left(x,x^{\prime }\right)=K_{\frac{1}{2}}^{M}\left(x,x^{\prime }\right)=\sigma_{p}^{2}e^{\left(-\frac{||x-x^{\prime }\left|\right|}{l}\right)}$$
and the case ν=+∞ gives the RBF as in Equation (Equation 5), e.g., $K^{RBF}=K_{+\infty }^{M}$. A trade-off between those two values is obtained for $\nu =\frac{3}{2}$ as also explained in [25]. The expression of the Matern kernel for $\nu =\frac{3}{2}$ is given by
(6)$$K_{\frac{3}{2}}^{M}\left(x,x^{\prime }\right)=\sigma_{p}^{2}\left(1+\frac{||x-x^{\prime }\left|\right|\sqrt{3}}{l}\right)e^{\left(-\frac{||x-x^{\prime }\left|\right|\sqrt{3}}{l}\right)}$$

All the hyperparameters $l,\sigma_{p}$ and the observation noise variance $\sigma_{n}$ (Equation (4)) can be set manually and/or obtained by maximum likelihood estimation (MLE) on a training data set. Note that products and sums of kernels are also kernels. This allows one to build customized kernels for a specific problem based on basic kernels such as those presented in this section.

The original GP regression problem considers the full set of training input and considers them as noise-free. Some recent work [26,27,28] proposes extensions for sparse GP regression and/or taking into account input uncertainties. Here, as we are only locally modeling the surface through the vertical sonar measurements, our training data are relatively few; thus, a sparse formulation is not mandatory. However, as our robot accumulates position uncertainties (due to dead-reckoning drift) during the scan acquisition, taking in account input uncertainties could significantly improve our estimation. In a first step, we currently assume noise-free inputs.

## 4. MSIS Scan Merging

### 4.1. Environment Model

We model the surface of the environment based on virtual vertical sonar measurements. More precisely, a 3D point on the surface is parameterized by the 6D pose of the robot and the vertical sonar rotation angle at which the point could have been seen by the vertical sonar. Formally, the surface is defined as a function which maps a 6D pose η and a scan rotation angle $\psi^{v}$ to an estimated distance $\rho^{v}$:
(7)$$f:\left\{\begin{array}{cc} SE(3)\times [-\pi ,\pi ] & \to R^{+}\\ \left(\eta ,\;\psi^{v}\right) & \mapsto \rho^{v} \end{array}\right.$$

This means that $\rho^{v}$ is the distance expressed in the vertical sonar plan (XY-plan in $\left\{S^{V}\right\}$) that would have been acquired for a measurement at the robot pose η and with the sonar rotation angle $\psi^{v}$. We cannot use here a terrain model of the form z=f(x,y) like in [21,29] as it is a non-functional relation in our karstic context.

As the robot trajectory is a 1D submanifold of SE(3), we can parameterize it with a simple scalar u∈R. We recall that kernels measure similarity between two inputs (cf. Section 3.4). If we consider a kernel K(.,.), we would like to have
(8)$$K\left(\eta_{1},\eta_{2}\right)=K\left(u_{1},u_{2}\right).$$


The first idea is to use the time *t* to describe the position of the trajectory manifold, as done in [25]. However, in this case Equation (Equation 8) is not verified. Indeed, the kernel length scale *l* as in (Equation 5) in the Gaussian process regression will be depending on time and not on space. The problem can be illustrated by considering an immobile robot at two instants $t_{1}$ and $t_{2}$ such that $t_{2}\gg t_{1}$ and $\eta \left(t_{1}\right)=\eta \left(t_{2}\right)$. We then have $K(t_{1},t_{2})\approx 0$ whereas $K\left(\eta_{1},\eta_{2}\right)=\sigma_{p}^{2}$.

To avoid this, we use the curvilinear abscissa as an input parameter based on the length of the trajectory in SE(3). We use the left-invariant Riemannian metric induced by the following point-independant Riemannian metric on GA(3) (general affine group) ([30], Section 3.2)
(9)$$\begin{array}{ccc} \forall A\in SE\left(3\right),\;\forall \;u,v\in T_{A}SE\left(3\right),\;{\langle u,v\rangle }_{A} & = & Tr(u^{T}v)\\ & = & {\langle A^{-1}u,A^{-1}v\rangle }_{e}\\ & = & {\langle s_{u},s_{v}\rangle }_{e}\\ & = & s_{u}^{T}Gs_{v} \end{array}$$
(10)$$\begin{array}{c} G=\left[\begin{array}{cc} I_{3} & 0\\ 0 & 2I_{3} \end{array}\right] \end{array}$$
where *e* is the identity element of SE(3), $T_{A}SE\left(3\right)$ the tangent plane at *A* and $s_{u},s_{v}\in \mathrm{se}\left(3\right)$.

The length of a curve η(t):R→SE(3) between two instants $t_{1}$ and $t_{2}$ is then given by
(11)$$L_{\eta (t1)}^{\eta (t2)}=\int_{t_{1}}^{t_{2}}{\langle \frac{d\eta }{dt},\;\frac{d\eta }{dt}\rangle }^{\frac{1}{2}}dt$$

The curvilinear abscissa *s* is simply given by
(12)$$\forall t,\;s\left(t\right)=L_{\eta (0)}^{\eta (t)}$$

In our work, we consider a discrete trajectory regularly updated from dead-reckoning fusion algorithms, fed with DVL and IMU measurements. Note that we only consider the trajectory during one full scan acquisition. The full trajectory estimation in a graph SLAM framework is out of the scope of this paper. Given the discrete trajectory $H=\{\eta_{i}\}$, we approximate the length as follows.
(13)$$\begin{array}{ccc} \forall i\in 〚2,n^{r}〛,\;s_{i}=L_{\eta_{1}}^{\eta_{i}} & = & \sum_{k=1}^{i}{\langle \Delta \eta_{k},\Delta \eta_{k}\rangle }^{\frac{1}{2}}\\ & = & \sum_{k=1}^{i}{\langle \eta_{k}⊖\eta_{k-1},\eta_{k}⊖\eta_{k-1}\rangle }^{\frac{1}{2}}\\ & = & \sum_{k=1}^{i}\sqrt{log{\left(\eta_{k}⊖\eta_{k-1}\right)}^{T}\;G\;log\left(\eta_{k}⊖\eta_{k-1}\right)}\\ & = & \sum_{k=1}^{i}\sqrt{log{\left(T_{k-1}^{-1}T_{k}\right)}^{T}\;G\;log\left(T_{k-1}^{-1}T_{k}\right)} \end{array}$$
where log is the logarithm map of SE(3) and $T_{k}\in GA\left(3\right)$ is the transformation matrix equivalent to the pose $\eta_{k}$.

Similarly to [25,29], our objective is to estimate *f* by training a Gaussian process on the vertical sonar measurements. The learned function is then used to estimate the most likely angles $\theta_{ij}^{h}$ inside the horizontal sonar beams measurements.

### 4.2. Algorithm Description

In this section, we provide a detailed description of our algorithm which is summarized in Algorithm 1.

First, we compute the curvilinear abscissa for each pose of the trajectory based on Equation (Equation 13) (line 1). The surface prior $f_{p}$ is then estimated by fitting an elliptic cylinder to the vertical sonar points (line 2). The Gaussian process (GP) modeling the environment surface is trained using the vertical sonar measurements and their corresponding curvilinear abscissa (line 3). Once done, for each horizontal sonar data, we estimate the most likely elevation angles $\theta_{ij}^{h}$ (line 5) and compute their corresponding uncertainties (line 7). In the following, we detail the process for each step.

#### 4.2.1. Environment Prior $f_{p}$

In this section, we explain how our environment prior $f_{p}$ is defined. Based on typical karstic environment, the natural choice is to use an elliptic cylinder as a prior shape. For a given pose η on the trajectory (equivalently its curvilinear abscissa) and a given sonar rotation angle $\psi^{v}$, $f_{p}\left(\eta ,\psi^{v}\right)$ returns the distance to the prior shape.

In order to fit this primitive on the vertical sonar measurements, we follow an approach similar to [31]. First, as for any non-linear optimization approach, we need a good initial shape. It is obtained by computing robust principal component analysis (RPCA) [32] on the vertical sonar point cloud. It provides three principals vectors which, in the case of a perfect elliptic cylinder, should correspond to the cylinder axis, the major and the minor elliptic axis. If the length of the estimated cylinder is higher than its width, then the eigenvector with the highest eigenvalue is the cylinder axis. However, we cannot make this assumption in our case. To find the cylinder axis among those principal vectors, we used the fact that every tangent on a cylinder surface is orthogonal to its axis. This means that by approximating tangent vectors by the vector formed by two successive points, their projections on the PCA bases should be minimal along the cylinder axis.

Then, the elliptic section is estimated by fitting the projections of all the points in the plane spanned by the two other principals axis [33]. The PCA base is then rotated around the cylinder axis to be aligned with the ellipse axis.

As can be seen with the orange elliptic cylinder in Figure 5, the dissymetric distribution of the vertical sonar points has a huge influence on the RPCA result. To further improve our estimation, we use the Levendberg–Marquardt (LM) algorithm to obtain the shape which minimizes orthogonal distances to the elliptic cylinder. The estimation vector $x\in R^{8}$ representing our elliptic cylinder is composed of a 6D pose in SE(3) and the scales along the minor and major ellipse axis. Here, we use on-manifold optimization on SE(3), as described in [22]. The final prior can be seen as the purple cylinder in Figure 5.

#### 4.2.2. Gaussian Process Regression and Kernel Choices

In this section, we explain the choice of kernels for the following Gaussian process regression. Based on our formulation of the surface through the functional *f* (cf. Section 4.1), we exploit its cylindrical structure by combining a kernel on the trajectory abscissa and a kernel on the sonar rotation angle:
(14)$$K_{prod}\left({\left[s\;\psi^{v}\right]}^{T},{\left[s^{\prime }\;\psi^{{}^{\prime }v}\right]}^{T}\right)=K_{s}\left(s,s^{\prime }\right)K_{\psi }\left(\psi^{v},\psi^{{}^{\prime }v}\right)$$

$K_{s}$ can be any kernel as proposed in Section 3.4. For $K_{\psi }$, as we deal with angles, we cannot use a simple euclidean distance. When choosing the metric to use, we have to ensure that the resulting kernel is positive definite [34,35]. We can define two distance $d_{chord}$ and $d_{geo}$ based respectively on the chordal distance and the geodesic distance on $S^{1}$:
(15)$$\begin{array}{ccc} d_{chord}\left(\psi ,\psi^{\prime }\right) & = & 2\sin\left(\frac{\psi -\psi^{\prime }}{2}\right) \end{array}$$
(16)$$\begin{array}{ccc} d_{geo}\left(\psi ,\psi^{\prime }\right) & = & acos\left(\cos\left(\psi -\psi^{\prime }\right)\right) \end{array}$$

In [35], it is shown that for geodesic distance $d_{geo}$ a $C^{2}$-Wendland function [36] can be used to define a valid kernel
(17)$$\forall 0\leq t\leq \pi ,\;W_{\pi }\left(t\right)=\left(1+\tau \frac{t}{\pi }\right){\left(1-\frac{t}{\pi }\right)}_{+}^{\tau },\;\tau \geq 4$$

For $d_{chordal}$, any kernel can be used.

In this paper, we only consider exponential $K^{exp}$ and Matern52 $K_{\frac{5}{2}}^{M}$ kernels with chordal distance for angles. In the experimental section, we investigate the effects of those different kernels and distances on the surface estimation.

#### 4.2.3. Maximum Likelihood Estimation of θ

Originally, we cannot obtain the 3D points observed by the horizontal sonar as any point in its beam may return the same measurement. We exploit the estimated surface by the vertical sonar to infer the elevation angles θ for each horizontal sonar measurements. The idea is that elevation angles corresponding to 3D points near the estimated surface are good estimate. This is illustrated in Figure 6. Hence, for each horizontal sonar measurement $\psi_{k}^{h},\rho_{k}^{h}$, we search for the elevation angles solutions of:
(18)$$\begin{array}{lll} \hat{\theta } & = & arg\max_{\theta \in \Theta^{H}}p\left({\hat{\rho }}^{v}|f,\psi_{k}^{h},\rho_{k}^{h},\eta_{\sigma_{h}\left(k\right)},\theta \right)\\ & = & arg\max_{\theta \in \Theta^{H}}ln\left(p\left({\hat{\rho }}^{v}|f,\psi_{k}^{h},\rho_{k}^{h},\eta_{\sigma_{h}\left(k\right)},\theta \right)\right)\\ & = & arg\max_{\theta \in \Theta^{H}}J\left(\theta ;\psi_{k}^{h},\rho_{k}^{h}\right) \end{array}$$
where ${\hat{\rho }}^{v}$ is the estimated range we would have obtained if this point was observed by the vertical sonar. In fact, depending on the environment configuration, several angles can correspond to one measure as previously illustrated in the Figure 4. Thus, instead of solving Equation (Equation 18), we search for local log-maximum likelihood values $\hat{\Theta }$:
(19)$$\hat{\Theta }=\left\{\hat{\theta }\in \Theta^{H}\;|\;\frac{\partial J}{\partial \theta }\left(\hat{\theta }\right)=0\;\mathrm{and}\;\frac{\partial^{2}J}{\partial \theta^{2}}\left(\hat{\theta }\right)<0\right\}$$

The corresponding algorithm is described in Algorithm 2 and detailed in the following.

As it is difficult to obtain a closed-form expression of $p\left({\hat{\rho }}^{v}|f,\rho_{k}^{h},\psi_{k}^{h},\eta_{\sigma^{h}\left(k\right)},\theta \right)$ relative to θ, we uniformly sample θ values in the range $\Theta^{H}$. In practice, $\Theta^{H}=\left[-\frac{b}{2},\;\frac{b}{2}\right]$ where *b* is the beam vertical aperture. We denote those samples $\theta_{q},q\in 〚1,n_{q}〛$. For each sample $\theta_{q}$, we compute the corresponding virtual 3D points $x(\theta_{q})$
(20)$$x\left(\theta_{q}\right)=g^{h}\left(\left[\begin{array}{ccc} \rho^{h} & \theta_{q} & \psi^{h} \end{array}\right],k\right)$$
with $g^{h}$ defined by Equation (Equation 1). To assess the likelihood of having measured this point on the surface, we need to use the previously trained GP model *f*. We therefore have to express this point in terms of *f* inputs and output. In other words, we need the curvilinear abscissa ${\hat{s}}_{q}$, the rotation angle ${\hat{\psi }}_{q}^{v}$ and range ${\hat{\rho }}_{q}^{v}$ as if the point was virtually seen by the vertical sonar. This is illustrated in Figure 7.

To obtain those parameters, we proceed as follows. First, let us denote $n_{\left|\{S^{V}\right\}}^{v}$ the rotation axis of the vertical sonar expressed in its local frame. It is also the normal vector of the sonar plane. We are searching a pose ${\hat{\eta }}_{q}$ along the trajectory such that $x(\theta_{q})$ belongs to the vertical sonar plane. To do so, we need to interpolate the robot trajectory. The first step is to determine in which segments of the trajectory ${\hat{\eta }}_{q}$ could belong to (several segments can be found). We recall that we note the global robot position by $\tau_{i}$ and its orientation by the quaternion $q_{i}$. The vertical sonar orientation quaternion is also defined by $q_{V}$. The rotation axis of the vertical sonar expressed in the global frame $n_{|\{0\},i}^{v}$ when the robot is in its *i*-th pose $\eta_{i}$ is given by
(21)$$n_{|\{0\},i}^{v}=R\left(q_{i}\right)R\left(q_{V}\right)n_{\left|\{S^{V}\right\}}^{v}$$

A segment (represented by indexes [i,i+1]) is considered if the projection of the vector $x\left(\theta_{q}\right)-\tau_{i}$ on the vertical rotation axis $n_{|\{0\},i}^{v}$ has a change of sign between the beginning and end of the segment (Figure 8). Indeed, thanks to the intermediate value theorem, such a segment contains the pose for which the projection is null. Formally, we consider the set $I_{q}$ of starting index such that
(22)$$I_{q}=\left\{i\;|\;\langle n_{|\{0\},i}^{v},x\left(\theta_{q}\right)-\tau_{i}\rangle \langle n_{|\{0\},i+1}^{v},x\left(\theta_{q}\right)-\tau_{i+1}\rangle <0\right\}$$

For each selected segment $\left[i,i+1\right],i\in I_{q}$, we denote $\eta_{\left[i,i+1\right]}:\left[0,1\right]\to SE\left(3\right)$ the linear interpolation between the poses $\eta_{i}$ and $\eta_{i+1}$. $\tau_{\left[i,i+1\right]}$ is defined by the first three coordinates of $\eta_{\left[i,i+1\right]}$ and is simply given by
(23)$$\tau_{\left[i,i+1\right]}\left(t\right)=\left(\tau_{i+1}-\tau_{i}\right)t+\tau_{i}$$

Let define the relative angular pose express in yaw–pitch–roll Euler angles between $\eta_{i}$ and $\eta_{i+1}$:
(24)$${\left[\begin{array}{ccc} \delta \psi & \delta \chi & \delta \phi \end{array}\right]}^{T}=Euler\left(\left(\eta_{i+1}⊖\eta_{i}\right)\left(3:7\right)\right)$$

We then have the associated interpolated rotation matrix δR(t):
(25)$$\delta R\left(t\right)=R\left(\delta \psi t,\delta \chi t,\delta \phi t\right)$$

The global rotation matrix R(t) is given by
(26)$$R\left(t\right)=R_{i}\delta R\left(t\right)$$

We can now define the linear interpolation of the vertical sonar rotation axis $n_{\left|\{0\},[i,i+1\right]}^{v}$:
(27)$$n_{\left|\{0\},[i,i+1\right]}^{v}\left(t\right)=R\left(t\right)R\left(q_{V}\right)n_{\left|\{S^{V}\right\}}^{v}$$

We search for the interpolation value $\hat{t}$ such that $x(\theta_{q})$ belongs to the sonar plane with normal $n_{\left|\{0\},[i,i+1\right]}^{v}\left(\hat{t}\right)$, that is
(28)$$\langle n_{\left|\{0\},[i,i+1\right]}^{v},x\left(\theta_{q}\right)-\tau_{\left[i,i+1\right]}\rangle \left(\hat{t}\right)=0$$

We assume small angle variation so that we can approximate δR to the first order
(29)$$\delta R\left(t\right)=\left[\begin{array}{ccc} 1 & -\delta \phi t & \delta \chi t\\ \delta \phi t & 1 & -\delta \psi t\\ -\delta \chi t & \delta \psi t & 1 \end{array}\right]$$

$\hat{t}$ is the root in [0,1] of the following order 2 polynomial expression
(30)$$n_{\left|\{S^{V}\right\}}^{v^{T}}R{\left(q_{V}\right)}^{T}R{\left(\hat{t}\right)}^{T}\left(\left(\tau_{i}-\tau_{i+1}\right)\hat{t}+x\left(\theta_{q}\right)-\tau_{i}\right)=0$$

We then have ${\hat{\eta }}_{q}=\eta_{\left[i,i+1\right]}\left(\hat{t}\right)$ the robot pose corresponding to the computed $\hat{t}$. We can now compute the curvilinear abscissa $\hat{s}$ corresponding to $\hat{t}$
(31)$${\hat{s}}_{q}=s_{i}+L_{\eta_{i}}^{{\hat{\eta }}_{q}}$$

To compute ${\hat{\psi }}_{q}^{v}$, we express $x(\theta_{q})$ in the vertical sonar frame coordinates corresponding to the robot pose ${\hat{\eta }}_{q,i}$
(32)$$\begin{array}{ccc} \hat{x}{\left(\theta_{q}\right)}_{\left|\{S^{V}\right\}} & = & x\left(\theta_{q}\right)⊖\left({\hat{\eta }}_{q}⊕p^{V}\right)={\left[\begin{array}{ccc} {\hat{x}}^{v} & {\hat{y}}^{v} & 0 \end{array}\right]}^{T} \end{array}$$
(33)$$\begin{array}{ccc} {\hat{\psi }}_{q}^{v} & = & atan2({\hat{y}}^{v},{\hat{x}}^{v}) \end{array}$$

Finally, the estimated range ${\hat{\rho }}_{q}^{v}$ is simply the L2 norm of $\hat{x}{\left(\theta_{q}\right)}_{\left|\{S^{V}\right\}}$
(34)$${\hat{\rho }}_{q}^{v}=\left|\right|\hat{x}{\left(\theta_{q}\right)}_{\left|\{S^{V}\right\}}{\left|\right|}_{2}$$

We can now compute the log-likelihood $J\left(\theta_{q};\psi_{k}^{h},\rho_{k}^{h}\right)=ln\left(p\left({\hat{\rho }}_{q}^{v}|f,\psi_{k}^{h},\rho_{k}^{h},\eta_{\sigma_{h}\left(k\right)},\theta_{q}\right)\right)$
(35)$$\begin{array}{ccc} \rho_{q}^{*,v} & ∼ & N({\bar{\rho }}_{q}^{*,v},\sigma_{\rho_{q}^{*,v}}^{2}) \end{array}$$
(36)$$\begin{array}{ccc} J(\theta_{q};\psi_{k}^{h},\rho_{k}^{h}) & = & -\left(\frac{1}{2\sigma_{\rho_{q}^{*,v}}^{2}}{\left({\bar{\rho }}_{q}^{*,v}-{\hat{\rho }}_{q}^{v}\right)}^{2}+ln\left(\sigma_{\rho_{q}^{*,v}}\right)+\frac{ln(2\pi )}{2}\right) \end{array}$$

The expressions for Equation (Equation 35) can be found in ([23], p17 Equation (2.25/2.26)) and are obtained using the GP regression trained to estimate *f* (Section 3.4).

#### 4.2.4. Estimation Uncertainty

In the previous section, we explained how the set $\hat{\Theta }$ is obtained. We are also interested in its related uncertainties. In other words, we want to define the probability distributions $p(\hat{\theta })$. A normal distribution is not adapted as its support is infinite while the width of our horizontal sonar beam is restricted to its aperture. A good choice is a scaled beta distribution [37] defined by two parameters α,β
(37)$$\hat{\theta }∼Beta\left(\alpha ,\beta ,\theta_{min},\theta_{max}\right)$$

In our case, we consider $\theta_{max}=-\theta_{min}=\frac{b}{2}$ where *b* is the horizontal sonar beam width. We then have the following probability density function
(38)$$f\left(\theta ;\alpha ,\beta ,b\right)=\frac{\Gamma (\alpha +\beta )}{\Gamma (\alpha )\Gamma (\beta )}\frac{{\left(\theta +\frac{b}{2}\right)}^{\alpha -1}{\left(\frac{b}{2}-\theta \right)}^{\beta -1}}{b^{\alpha +\beta -1}}$$
where Γ is the gamma function which extends the factorial function to complex numbers and is defined by
(39)$$\forall z\in C,\;\Gamma \left(z\right)=\int_{0}^{\infty }x^{z-1}e^{-x}dx,\;Re\left(z\right)>0$$

For each estimated $\hat{\theta }$ from the previous section, the parameters α,β can be computed from a given mean $\bar{\theta }$ and variance $\sigma_{\theta }^{2}$:
(40)$$\begin{array}{c} \alpha =\left(b+2\bar{\theta }\right)\frac{b^{2}-4\left({\bar{\theta }}^{2}+\sigma_{\theta }^{2}\right)}{8b\sigma_{\theta }^{2}} \end{array}$$
(41)$$\begin{array}{c} \beta =\left(b-2\bar{\theta }\right)\frac{b^{2}-4\left({\bar{\theta }}^{2}+\sigma_{\theta }^{2}\right)}{8b\sigma_{\theta }^{2}} \end{array}$$

In our case, the mean is simply the estimated θ ie $\bar{\theta }=\hat{\theta }$. The variance $\sigma_{\theta }^{2}$ requires more computation and is estimated as the inverse of the Fisher information $I(\bar{\theta })$:
(42)$$\sigma_{\theta }^{2}=\frac{1}{I(\bar{\theta })}$$
(43)$$\begin{array}{ccc} I(\theta ) & = & -E\left(\frac{\partial^{2}ln\left(p(\rho^{v}|\theta )\right)}{\partial \theta^{2}}|\theta \right)\\ & = & -\int \frac{\partial^{2}ln(p\left(\rho^{v}|\theta \right)}{\partial \theta^{2}}p\left(\rho^{v}|\theta \right)dr \end{array}$$


Direct computation of Equation (43) gives us (see Appendix A for the derivation):
(44)$$\begin{array}{c} \sigma_{\theta }^{2}=\frac{\sigma_{\rho^{v}}^{2}\left(\bar{\theta }\right)}{2{\left({\frac{\partial \sigma_{\rho^{v}}}{\partial \theta }}_{|\bar{\theta }}\right)}^{2}+{\left({\frac{\partial \bar{\rho^{v}}}{\partial \theta }}_{|\bar{\theta }}\right)}^{2}} \end{array}$$

Note that a uniform distribution can still be represented by a beta distribution with shape parameters α=β=1. Indeed, from Equations (Equation 39) and (Equation 38) we have
(45)$$f\left(\theta ;1,1,b\right)=\frac{\Gamma (2)}{b\Gamma (1)\Gamma (1)}=\frac{1}{b}$$
which is the density probability for a uniform distribution on $\left[-\frac{b}{2},\frac{b}{2}\right]$.

We now define a threshold $\sigma_{l}$ to filter out variances $\sigma_{\theta }$ leading to invalid Beta distribution parameters . First, the parameters α,β of a Beta distribution must be strictly positive:
(46)$$\alpha \geq 0,\beta \geq 0\Leftrightarrow \sigma_{\theta }^{2}\leq {\left(\frac{b}{2}\right)}^{2}-{\bar{\theta }}^{2}$$

Furthermore, as we consider each distribution as unimodal, we must have α≥1 or β≥1. (In the case α,β<1, the beta distribution is bimodal with modes at $\left\{-\frac{b}{2},\frac{b}{2}\right\}$. A mode corresponds to a local maxima of a probability density function (pdf).) From the expressions (Equation 40) and (Equation 41), we have
(47)$$\begin{array}{ccc} \alpha \geq 1\Leftrightarrow \sigma_{\theta }^{2} & \leq & A\left(\bar{\theta }\right)=\frac{{\left(b+2\bar{\theta }\right)}^{2}\left(b-2\bar{\theta }\right)}{4\left(3b+2\bar{\theta }\right)} \end{array}$$
(48)$$\begin{array}{ccc} \beta \geq 1\Leftrightarrow \sigma_{\theta }^{2} & \leq & B\left(\bar{\theta }\right)=A\left(-\bar{\theta }\right) \end{array}$$

This implies
(49)$$\sigma_{\theta }^{2}\leq \sigma_{l}^{2}\left(\bar{\theta }\right)=max\left(A\left(\bar{\theta }\right),A\left(-\bar{\theta }\right)\right)=A(|\bar{\theta }\left|\right)$$

It is also easy to verify that the first constraint α,β≥0 is included in the second as we have
(50)$$\forall \bar{\theta }\in \left[-\frac{b}{2},\frac{b}{2}\right],\;\sigma_{l}^{2}\left(\bar{\theta }\right)=A(|\bar{\theta }\left|\right)\leq {\left(\frac{b}{2}\right)}^{2}-{\bar{\theta }}^{2}$$

We also add another threshold $\sigma_{m}$ to filter out valid but too uncertain estimations. It is defined by
(51)$$k\sigma_{m}=\frac{b}{2}$$

The parameter *k* controls the filter sensibility, with higher values being more restrictive. In our implementation, we used a default value k=3. The final threshold $\sigma_{T}$ is then
(52)$$\begin{array}{c} \sigma_{T}^{2}\left(\bar{\theta }\right)=min\left(\sigma_{l}^{2}\left(\bar{\theta }\right),\sigma_{m}^{2}\right) \end{array}$$

Figure 9 shows the graphs of $\sigma_{l},\sigma_{m}$ and $\sigma_{T}$.

Figure 10 illustrates the uncertainty estimation process. The colors used to represent the beta distributions directly on the measurement arc are obtained through the following mapping
(53)$$m:\left\{\begin{array}{cc} R^{+} & \to [0,1]\\ x & \mapsto \frac{2}{\pi }arctan\left(bx\right) \end{array}\right.$$

Note that the central color (turquoise) is obtained for $x=\frac{1}{b}$, which is the pdf for a uniform distribution.

**Algorithm 1** Scan merging.**Input:** Sonar measurements $M^{h}$, $M^{v}$, Robot trajectory *H***Output:** Estimated horizontal measurements $p(\hat{\theta })$  1: $s={\left[\begin{array}{ccc} s_{1} & \cdots & s_{n_{r}} \end{array}\right]}^{T}\leftarrow s\left(H\right)$▹ Curvilinear abscissa, Equation (Equation 12)  2: $f_{p}\leftarrow fitEllipticCylinder\left(M^{v}\right)$▹ Surface prior  3: $f\leftarrow \mathrm{GP}(s,\;M^{v},f_{p})$▹Surface Estimation by Gaussian Process  4: **for each**
$\psi_{i}^{h},\rho_{ij}^{h}$
**do**▹ Estimation  5:    $\hat{\Theta }=\mathrm{ElevationAngleMLE}\left(f,s,\psi_{i}^{h},\rho_{ij}^{h}\right)$▹ See Algorithm 2  6:    **for**
$\hat{\theta }\in \hat{\Theta }$
**do**  7:        $p\left(\hat{\theta }\right)\leftarrow \mathrm{UncertaintyEstimate}\left(p\left(\rho^{v}|\theta \right),\hat{\theta }\right)$  8:    **end for**  9: **end for**

**Algorithm 2** Elevation angle MLE.**Input:** Trained surface GP *f*, a horizontal sonar measure $\left(\psi_{k}^{h},\rho_{k}^{h}\right)$, robot trajectory *H* and the corresponding curvilinear abscissa s, Rotation axis of vertical scan expressed in the vertical sonar frame $n_{\left|\{S^{V}\right\}}^{v}$, Number of θ samples $n_{q}$**Output:** Estimated local maximum elevation angles $\hat{\Theta }$1:$\hat{\Theta }\leftarrow \emptyset$, $\Delta_{\theta }\leftarrow \frac{\theta_{max}-\theta_{min}}{n_{q}}$2:
**for**
*q*
**do**
3:    $\theta_{q}\leftarrow \theta_{q-1}+\Delta_{\theta }$4:    $x\leftarrow g^{h}\left(\left[\begin{array}{ccc} \rho_{k}^{h} & \theta_{q} & \psi_{k}^{h} \end{array}\right],k\right)$▹ Equation (Equation 20)5:    I←∅6:    **for**
$i\in [1,n^{r}-1]$
**do**7:        $n_{|\{0\},i}^{v},n_{|\{0\},i+1}^{v}\leftarrow R\left(q_{i}\right)R\left(q_{V}\right)n_{\left|\{S^{V}\right\}}^{v},\;R\left(q_{i+1}\right)R\left(q_{V}\right)n_{\left|\{S^{V}\right\}}^{v}$▹ Equation (Equation 21)8:        **if**
$\langle n_{|\{0\},i}^{v},x-\tau_{i}\rangle \langle n_{|\{0\},i+1}^{v},x-\tau_{i+1}\rangle <0$**then**▹ Equation (Equation 22)9:           I←i10:        **end if**11:    **end for**12:    ${\hat{\theta }}_{q}\leftarrow 0$13:    $l_{q}^{max}\leftarrow 0$▹ Maximum likelihood value14:    **for**
i∈I
**do**15:        $n_{\left|\{0\},[i,i+1\right]}^{v}\left(t\right)\leftarrow linearInterp\left(n_{|\{0\},i}^{v},\;n_{|\{0\},i+1}^{v}\right)$16:        $\tau_{\left[i,i+1\right]}\left(t\right)\leftarrow linearInterp\left(\tau_{i},\;\tau_{i+1}\right)$17:        $\eta_{\left[i,i+1\right]}\left(t\right)\leftarrow linearInterp\left(\eta_{i},\;\eta_{i+1}\right)$18:        $\hat{t}\leftarrow \mathrm{root}\left(\langle n_{\left|\{0\},[i,i+1\right]}^{v},x-\tau_{\left[i,i+1\right]}\rangle \left(t\right)\right)$▹ Equations (Equation 28) and (Equation 30)19:        ${\hat{\eta }}_{q}\leftarrow \eta_{\left[i,i+1\right]}\left(\hat{t}\right)$20:        ${\hat{x}}_{\left|\{S^{V}\right\}}={\left[{\hat{x}}_{i}^{v}\;{\hat{y}}_{i}^{v}\;0\right]}^{T}\leftarrow x⊖\left({\hat{\eta }}_{q}⊕p^{V}\right)$▹ Equation (Equation 32)21:        ${\hat{s}}_{q}\leftarrow s_{i}+L_{\eta_{i}}^{{\hat{\eta }}_{q}}$▹ Equation (Equation 31)22:        ${\hat{\psi }}_{q}^{v}\leftarrow atan2\left({\hat{y}}^{v},{\hat{x}}^{v}\right)$▹ Equation (Equation 33)23:        ${\hat{\rho }}_{q}^{v}\leftarrow \left|\right|{\hat{x}}_{\left|\{S^{V}\right\}}{\left|\right|}_{2}$▹ Equation (Equation 34)24:        $l_{q}\leftarrow ln\left(p\left({\hat{\rho }}_{i}^{v}|f,{\hat{s}}_{i},{\hat{\psi }}_{i},\theta_{q}\right)\right)$▹ Equation (Equation 36)25:        **if**
$l_{q}>l_{q}^{max}$
**then**26:           $l_{q}^{max}\leftarrow l_{q}$27:           ${\hat{\theta }}_{q}\leftarrow \theta_{q}$28:        **end if**29:    **end for**30:    **if**
$\left(l_{q}^{max}<l_{q-1}^{max}\right)\wedge \left(l_{q-1}^{max}>l_{q-2}^{max}\right)$
**then**▹ $\frac{\partial^{2}J}{\partial \theta^{2}}<0$, Equation (Equation 19)31:        $\hat{\Theta }\leftarrow {\hat{\theta }}_{q}$32:    **end if**33:
**end for**


**Algorithm 3** Uncertainty estimate.**Input:** Likelihood $p(\rho^{v}|\theta )$, local maximum likelihood $\bar{\theta }$, beam width *b***Output:** Posterior $p(\hat{\theta })$  1: $I\left(\bar{\theta }\right)\leftarrow \frac{2{\left(\frac{\partial \sigma_{\rho }}{\partial \theta }\right)}^{2}+{\left(\frac{\partial \bar{\rho }}{\partial \theta }\right)}^{2}}{\sigma_{\rho }^{2}}$▹ Fisher information Equation (Equation 43)  2: $\sigma_{\theta }^{2}\leftarrow \frac{1}{I(\bar{\theta })}$  3: α,β←1▹ Beta distr. parameters  4: **if**
$\sigma_{\theta }\leq \sigma_{T}$
**then**▹ Equation (Equation 52)  5:    $\alpha \leftarrow \left(b+2\bar{\theta }\right)\frac{b^{2}-4\left({\bar{\theta }}^{2}+\sigma_{\theta }^{2}\right)}{8b\sigma_{\theta }^{2}}$  6:    $\beta \leftarrow \left(b-2\bar{\theta }\right)\frac{b^{2}-4\left({\bar{\theta }}^{2}+\sigma_{\theta }^{2}\right)}{8b\sigma_{\theta }^{2}}$  7: **end if**  8: $p\left(\hat{\theta }\right)\leftarrow Beta\left(\alpha ,\beta ,-\frac{b}{2},\frac{b}{2}\right)$

## 5. Experimental Results

To validate the proposed method described previously, we made experiments on simulated environments shown in Figure 11. Simulation allows to modify the experimental configuration such as the measurements density or the smoothness of the environment surface. It also provides a ground truth so that quantitative analysis can be carried out.

We are currently working on a novel version of pICP adapted to the presented methodology. It will be integrated in a pose-graph SLAM framework and implemented on the Telemaque AUV shown in Section 3.2. Real experiments will then be performed in the Lez karst aquifer in the South of France [38,39].

### 5.1. Implementation

Our current implementation consists of a simulation engine implemented in C++ based on the MRPT library [40]. The Gaussian process regression is done in python using the GPy library [41]. In this paper, we do not focus on the algorithm complexity and computation time. It will be part of our future work as a mandatory step towards its use in real applications. Note that final implementation is planned to be fully C++ using the Limbo library [42] for the Gaussian process regression.

### 5.2. Settings

We present here the settings for which we test our algorithm. We consider three variables which are the surface smoothness, the measure density and the kernel choices.

We can also include the robot trajectory as a fourth variable. If not explicitly stated, we consider a straight linear robot trajectory as it results in clearer display easier to understand, especially for qualitative analysis. Quantitative results are also provided for random trajectories.

To apprehend the effect of environment surface smoothness on the surface estimation, we also evaluate our approach on a chaotic model where we have added some random noise to a cylindrical karst model. We also consider two measurement densities. While fixing the robot movement at each simulation step, we generate dense measurements by considering a large karst (radius of 22m) and sparse measurements with a scaled down karst (radius of 1m). Note that the dense case is the similar to the general density of measurements expected in real karst exploration. The sonar measurements, used as training data for the surface estimation, are computed using ray-tracing on the environment mesh.

For the kernels, we consider the products between kernels on trajectory abscissa $K_{s}\in \left\{K^{exp},K_{\frac{3}{2}}^{M},K_{\frac{5}{2}}^{M},K^{RBF}\right\}$ and kernels on the sonar rotation angle $K_{\psi }\in \left\{K^{exp}\circ d_{chordal},K_{\frac{5}{2}}^{M}\circ d_{chordal}\right\}$. This is summarized in Table 1.

### 5.3. Surface Estimation

#### 5.3.1. Effect of Kernel Choices

In order to quantitatively assess the performance of our algorithm, we have modeled our karst environment using the 3D modeling software Blender (http://www.blender.org). Unless otherwise stated, we optimize the GP with all the kernel parameters presented in Section 3.4. We discuss when needed the benefit and/or effect from fixing some of those parameters. As we are dealing with surface in 3D, it is hard to have a clear representation of the GP results. Figure 12 shows some examples of estimated surface with the mean (green) and the lower/upper bound at −/+3σ (dark blue/blue). To assess the effect of kernels on the surface estimation, we represent the surface estimation by considering upper contours in the XY, XZ and ZY planes (The selected ZY plane is set at the middle of the trajectory).

One should choose kernels which provides the best trade-off between confidence (variance) in the estimation and accuracy. Ideally, the variance in the surface estimation should be high enough to contain the real surface in its confidence interval. However, too high variance will lead to uniform distribution for the elevation angle estimation.

Note that as our surface estimation operates per full scan and thus is local, one can adapt dynamically the kernels used during operation depending on a priori information on the robot environment.

##### 
Kernel $K_{s}$


We fix here $K_{\psi }=K_{\frac{5}{2}}^{M}\circ d_{chordal}$ and consider the cases where the lengthscale $l_{\psi }$ is learned or fixed to 0.5. The objective here is to apprehend the behavior of each kernel $K_{s}$. Figure 13a (resp. Figure 13b) shows the Gaussian process regression results in the sparse and dense configuration for $l_{\psi }=0.5$ (resp. learned $l_{\psi }$ in Table 2b). From those graphics, we see that when $l_{\psi }=0.5$, the uncertainties in the surface estimations are different in the sparse and dense cases. In the former, we have by increasing order of uncertainty the Matern32, Matern52, rbf and exponential kernels. In the latter, we have Exponential, Matern32, Matern52 and RBF. Based solely on kernels smoothness, we expect the results seen in the dense case. This shows that the kernels behavior depends heavily on the environment configuration. We explain this difference by the bigger variations in measured ranges in the sparse case compared to the dense case (Figure 12). Obviously, the accuracy of the estimated surface is worse in the sparse case as large part of the surface are unseen by the vertical sonar.

When the $l_{\psi }$ is learned, we see results similar between the sparse and dense case. The variability of the results obtained shows the importance to consider both kernels $K_{s}$ and $K_{\psi }$ simultaneously.

##### 
Kernel $K_{\psi }$


We now fix $K_{s}=K_{\frac{5}{2}}^{M}$. Figure 14a,b show the results obtained when choosing the $K^{exp}\circ d_{chordal}$ or $K_{\frac{5}{2}}^{M}\circ d_{chordal}$ kernel for different fixed lengthscales. Table 2 contains the values of lengthscale $l_{\psi }$ obtained when learning them. We observe that the exponential kernel gives lower variance than required to take into account for the unknown environment. The first row of Table 2 shows that we obtain very high values for learned lengthscale. It shows that the GP failed to find appropriate value for this parameter. On the other hand, the kernel based on Matern52 gives more appropriate uncertainties controlled by the lengthscale parameter. The learned lengthscales are in this case more consistent.

The main conclusion of those results is that the choice of $K_{s}$ is not obvious. In other works dealing with surface estimation [21,25], $K_{\frac{5}{2}}^{M}$ and $K_{\frac{5}{2}}^{M}$ were chosen as being a compromise between $K^{exp}$ and $K^{RBF}$. However, depending on the configuration, it could be interesting to consider other kernels.

Concerning $K_{\psi }$, $K_{\frac{5}{2}}^{M}\circ d_{chordal}$ seems to be the best choice. Note that we are not exhaustive on the tested kernels. We use here generic kernel used extensively in the machine learning field. It would be interesting to further investigate on other kernels or creation of customized kernels adapted to the task at hand.

From the previous results, the pair of kernels $K_{s}=K_{\frac{5}{2}}^{M},K_{\psi }=K_{\frac{5}{2}}^{M}\circ d_{chordal}$ seems to be a reasonable choice and we use them in the following experiments.

We also tested our surface estimation approach on a more chaotic karst environment as shown in Figure 15.

#### 5.3.2. Random Trajectories

In the previous sections, we only considered straight line trajectory to facilitate the visualization. We present here the surface estimation for a robot with a randomly generated trajectory. This trajectory is a simple translation along x with an added Gaussian noise. It is illustrated in Figure 16a. Figure 16 shows the corresponding surface estimation. Note that the estimated surface cannot be directly compared with results in the previous section: the trajectories being differents, the points observed by the vertical sonar are different.

### 5.4. Evaluation of Elevation Angle Estimation

In this section, we aim at quantitatively assessing the proposed approach. We simulated random trajectories in modeled karsts in the following configurations: smooth karst with sparse/dense measurements and chaotic karst with dense measurements. The surfaces are estimated using the method described previously. Each simulated horizontal sonar beams is sampled by considering the ranges measured at 5 equally spaced elevation angles. Note that each of those samples corresponds to an arc of measure for which we estimate the elevation angle distribution as represented in Figure 17. For the robot trajectories, we used a RNG generator with a fixed seed for simulation in a given environment. It means that for the sparse, dense and chaotic cases each trial with a different kernel is based on the same randomly generated trajectory. However, note that as the estimated surface depends on the kernel used in the GP regression, the number of non-uniform estimated distributions will vary.

The performance of our algorithm is difficult to measure directly. One possible metric would be to compute the likelihood of the true elevation angle relative to the estimated distribution. However, as discussed in Section 3.3, one range can correspond to different real points and hence to different distributions. We choose to measure the error of an estimated non-uniform distribution (A non-uniform distribution arises when the beam measurement concerns an uncertain part of the estimated surface.) as the distance expectation from the measurement beam to the environment mesh. We denote d(θ) the distance from a point at elevation angle θ to the surface mesh and call this expected error the posterior error $E_{post}\left(d\right)$. We compare it to the prior error $E_{prior}\left(d\right)$ which is simply the distance expectation of the same measurement beam with a uniform distribution. Formally those errors are defined as follows
(54)$$\begin{array}{ccc} E_{prior}\left(d\right) & = & \int_{-\theta_{m}}^{\theta_{m}}d\left(\theta \right)p\left(\theta \right)d\theta =\frac{1}{2\theta_{m}}\int_{-\theta_{m}}^{\theta_{m}}d\left(\theta \right)d\theta \end{array}$$
(55)$$\begin{array}{ccc} E_{post}\left(d\right) & = & \int_{-\theta_{m}}^{\theta_{m}}d\left(\theta \right)p\left(\theta |f\right)d\theta \end{array}$$

In practice, those expectations are numerically approximated using the same $n_{q}$ samples $\theta_{q}$ defined in Section 4.2.3 so that
(56)$$\begin{array}{ccc} E_{prior}\left(d\right) & \approx & \frac{1}{n_{q}}\sum_{q=1}^{n_{q}}d\left(\theta_{q}\right) \end{array}$$
(57)$$\begin{array}{ccc} E_{post}\left(d\right) & \approx & \frac{\sum_{q=1}^{n_{q}}d\left(\theta_{q}\right)p\left(\theta_{q}|f\right)}{\sum_{q=1}^{n_{q}}p\left(\theta_{q}|f\right)} \end{array}$$

In the following experiments, we use $n_{q}=200$ and beam width b=35 deg. For each fixed settings (smooth karst with sparse and dense measurements, chaotic karst), we compute the errors for several kernels $K_{s}$ while fixing $K_{\psi }=K_{\frac{5}{2}}^{M}\circ d_{chordal}$. Error distributions are represented using box-and-whisker diagram. The rectangular box extends from the lower Q1 to the upper quartile Q3 of the distribution. The filled (resp. dot) line corresponds to the median (resp. mean). The upper whisker is drawn above the upper quartile up to the largest value at a distance below 1.5 times the interquartile range IQR Q3−Q1 (It corresponds to the rectangular box vertical length.). Similarly, the lower whisker is drawn at the lowest value value above 1.5 times the IQR. Values above and below the whiskers are drawn as circles. For each kernel $K_{s}$, the non-uniformly distributed beams considered for error computation are different. Indeed, as the surface estimation varies, some beam are estimated as uniform distribution for some kernel but not for others. Hence the need to compute the prior error for each kernel. The results are represented by pair of corresponding diagrams in Figure 18. We also compare the number of non-uniform distributions estimated with each kernel in Figure 18d. Table 3 provides the values of medians and means for each case.

We globally observe that in all cases regardless to the kernel considered our approach improves significantly upon the prior configuration. In particular, we have a relatively high percentage of the distribution with low errors. Besides, the number of estimated non-uniform distributions can vary depending on the kernel used. In our context, the estimated distributions are expected to be further processed by higher level algorithms to reject outliers.

In the sparse case (Figure 18a), the intermediate kernels Matern32 and Matern52 give the best results. We notice some discrepancy in the number of relevant estimations (non-uniform distribution) as can be seen in Figure 18d. This relates to results observed in Table 2a,b where the surface estimated with the exponential Kernel has less uncertainty. In this particular case, this kernel gives an overconfident surface estimation.

In the dense case (Figure 18b), the Matern32 and Matern52 kernels also give the best results with the RBF kernel being more erroneous. There is also a drop in the number of non-uniform distributions obtained for the RBF kernel.

Finally, Figure 18c shows the results obtained for a chaotic karst. Note that in this case, the difference between the prior and posterior are relatively small compared to the previous results (beam measurements are generally intersecting the environment several times). Even in this more difficult configuration, an important proportion of the estimated distribution gives low distance expectation error.

Overall those different observations validates what was theoretically expected. In particular, it also justifies the use of Matern52 kernel for $K_{s}$ as the best default choice. Our experiments is however limited to non exhaustive simulation results. Those results are thus a guideline to better understand the effect of kernel choices and this will be refined based on futur experiments on real data.

## 6. Conclusions

In this paper, we presented a method for probabilistic sonar scan merging in the context of underwater karstic exploration. It consists of using the measurements of a high resolution sonar (narrow beam) to estimate the measurement of another lower resolution sonar (wide beam). This is done using the cofounding variable linking both sonar measurements which is the environment. The first sonar measurements are used to obtain a probabilistic approximation of the surrounding surface through Gaussian process regression. It was then leveraged to estimate the probability distribution of the second sonar measurements.

In this work, we validated our approach to obtaining a probabilistic estimate of elliptic cylindrical shaped surface such as a karst aquifer. We analyzed the behavior of different kernel choices on the surface estimation. The results obtained were generally in agreement with the smoothness of considered kernels, but some interesting differences were also noted, notably between the sparse and dense measurements case. We also showed quantitative results based on the expected distances between measurement beams and the simulated mesh. We compared the distances obtained with our estimated elevation angles and a uniformly distributed prior distribution. On our simulated smooth karst, we improved the median expected distance by a ratio of 6.1 in the dense measurement case which corresponds to the measurements expected in real experiments and similar to those obtained in the dataset provided by [19]. We also show significant improvements in more difficult settings with a ratio of 4.1 for sparse measurement in a smooth karst and a ratio of 1.62 on a difficult chaotic environment.

This paper aimed at validate our approach and we will focus on improving it in our future works. There are still areas of improvement to explore. We are currently not taking in account uncertainties on the robot trajectory during a full scan acquisition and the noise on the sonar measurements. Besides, we can also improve the GP regression computation time using sparse GP based on inducing points. Another direction is to develop non-stationary kernel (Kernel which does not solely depends on the distance between its two inputs. Informally, it can modeled with variable lengthscale.) as proposed in [29]. It could be particularly interesting to deal with an environment containing strong discontinuities.

The proposed methodology is only a first step towards a real application for underwater karstic exploration. We are currently working on its usage by adapting probabilistic ICP [16] to our estimated point distributions inside a pose graph SLAM framework.

## Figures and Tables

**Figure 1 sensors-20-04028-f001:**
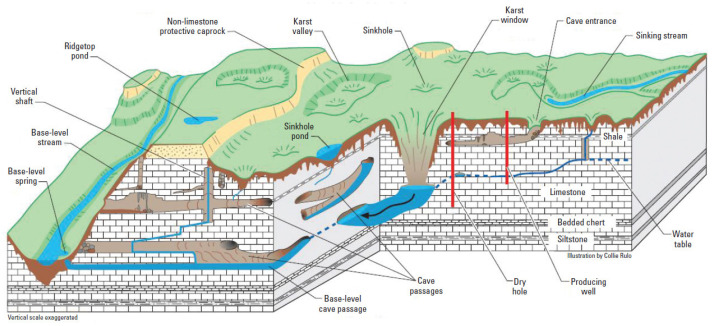
Karstic environment (extracted from [5]).

**Figure 2 sensors-20-04028-f002:**
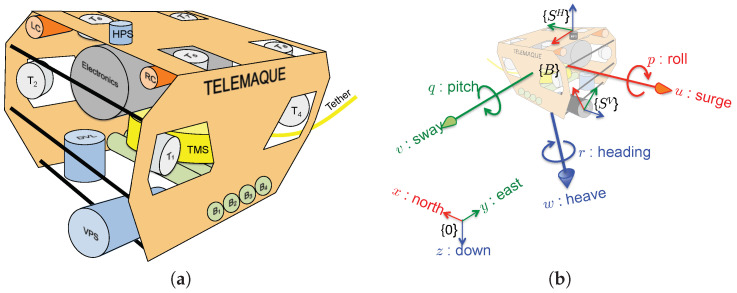
(**a**) Model of the underwater robot Telemaque, equipped for karstic exploration, and (**b**) frames definition.

**Figure 3 sensors-20-04028-f003:**
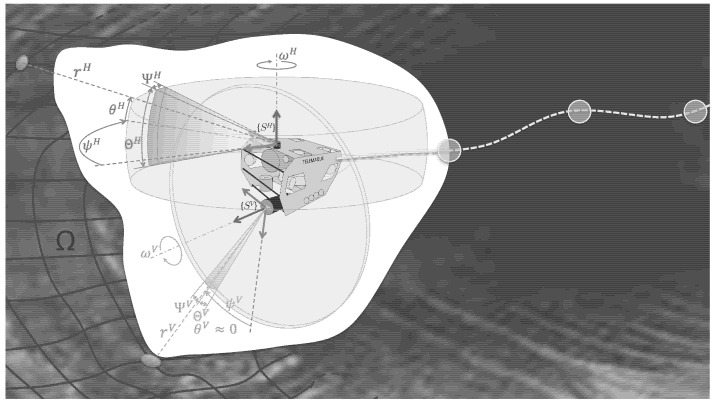
The vertical and horizontal profiling sonars mounted on Telemaque while scanning the environment’s surface Ω.

**Figure 4 sensors-20-04028-f004:**
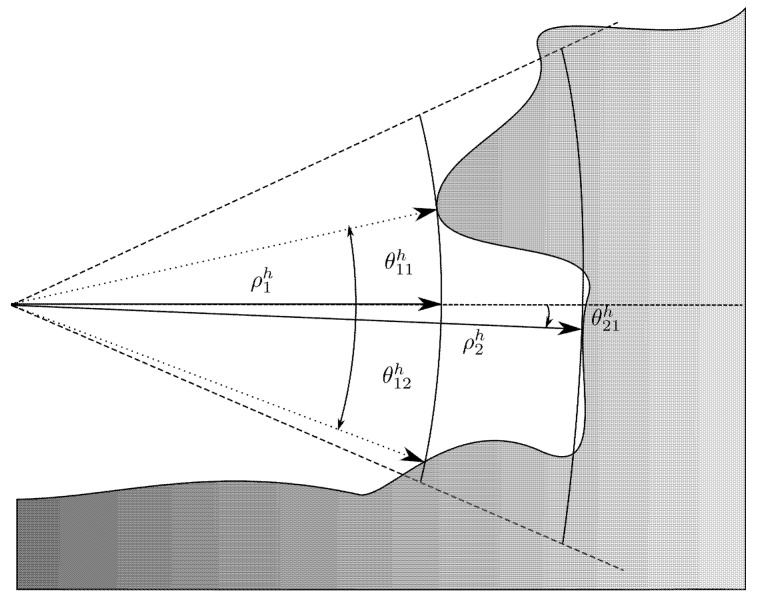
A single measure from the horizontal sonar can correspond to several 3D points and in consequence to several elevation angles.

**Figure 5 sensors-20-04028-f005:**
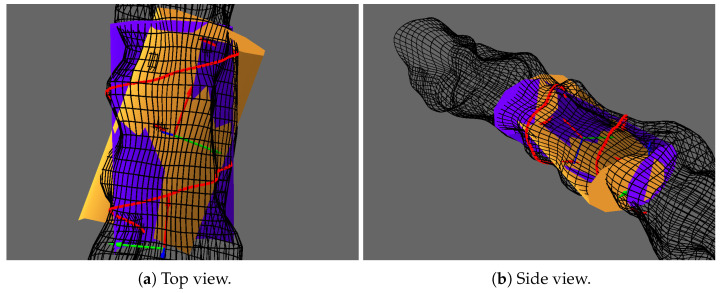
Elliptic cylinder fitting of vertical sonar points for the prior shape illustrated from Top (**a**) and Side (**b**) views. The RPCA results are represented in orange and the final results after Levendberg–Marquardt optimization in purple. The RPCA base is represented by a frame in the center of the cylinder.

**Figure 6 sensors-20-04028-f006:**
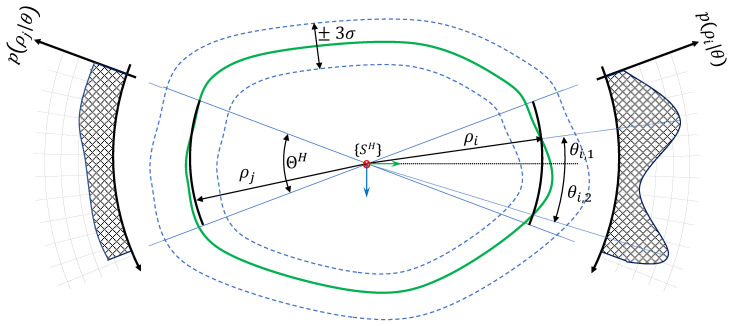
Illustration of elevation angle maximum likelihood estimation on a vertical slice. The green curve is the mean surface (cofounded here with the true surface for visibility) and blue curves are the confidence interval at ±3σ. Note that in this particular configuration $\rho =\rho^{h}={\hat{\rho }}^{v}$.

**Figure 7 sensors-20-04028-f007:**
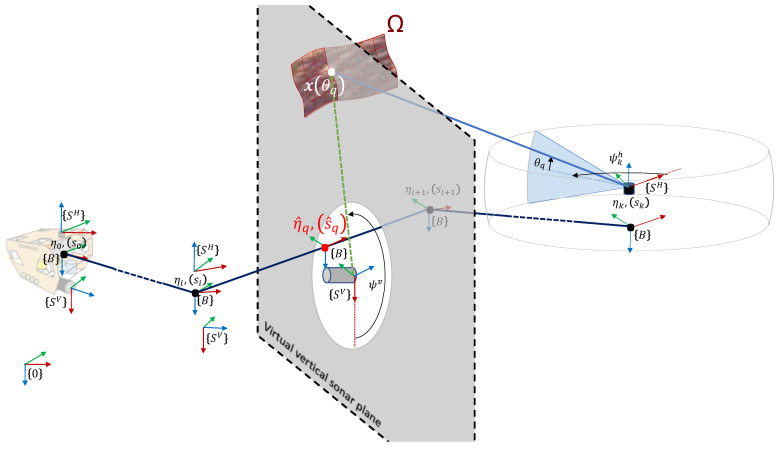
Elevation angle estimation illustration where the vertical sonar plane is parallel to the ZY plane of the robot.

**Figure 8 sensors-20-04028-f008:**
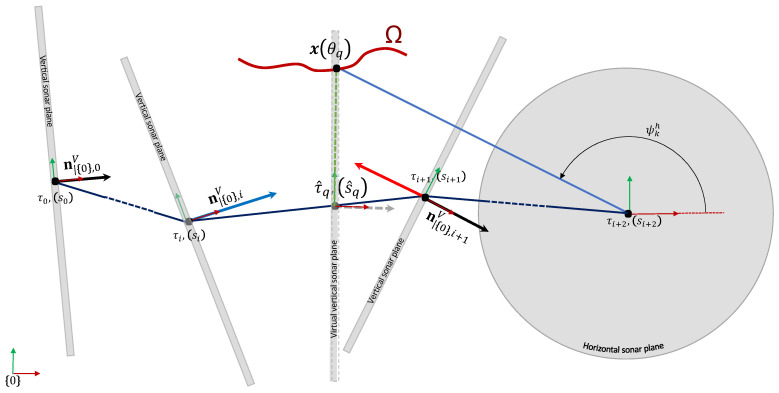
Segment selection for the *k*-th horizontal sonar measurement and *q*-th sample $x(\theta_{q})$. We have represented the vertical sonar planes and rotation axis $n_{|\{0\},i}^{v}$ along the trajectory from a top view. The blue (resp. red) vector corresponds to $\langle n_{|\{0\},i}^{v},x\left(\theta_{q}\right)-\tau_{i}\rangle n_{|\{0\},i}^{v}$ (resp. $\langle n_{|\{0\},i+1}^{v},x\left(\theta_{q}\right)-\tau_{i+1}\rangle n_{|\{0\},i+1}^{v}$).

**Figure 9 sensors-20-04028-f009:**
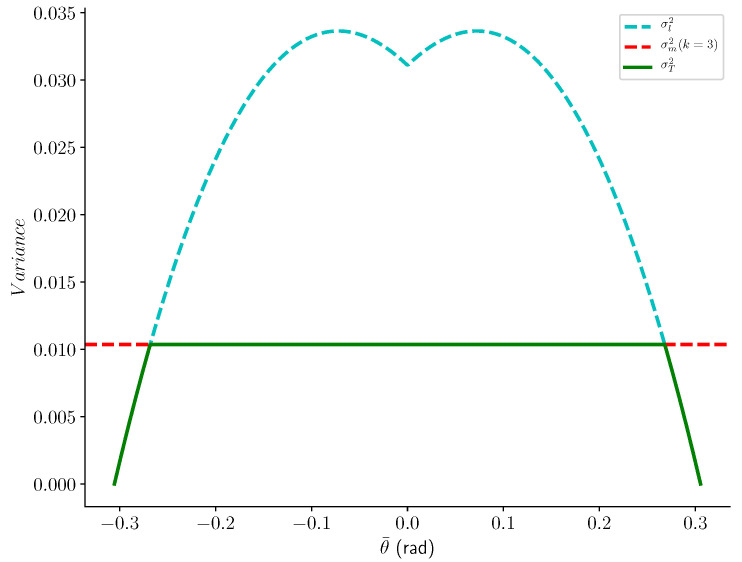
Thresholds $\sigma_{l}$, $\sigma_{m}$ and $\sigma_{T}$ relative to $\bar{\theta }$ with b=0.61 radians (35 deg).

**Figure 10 sensors-20-04028-f010:**
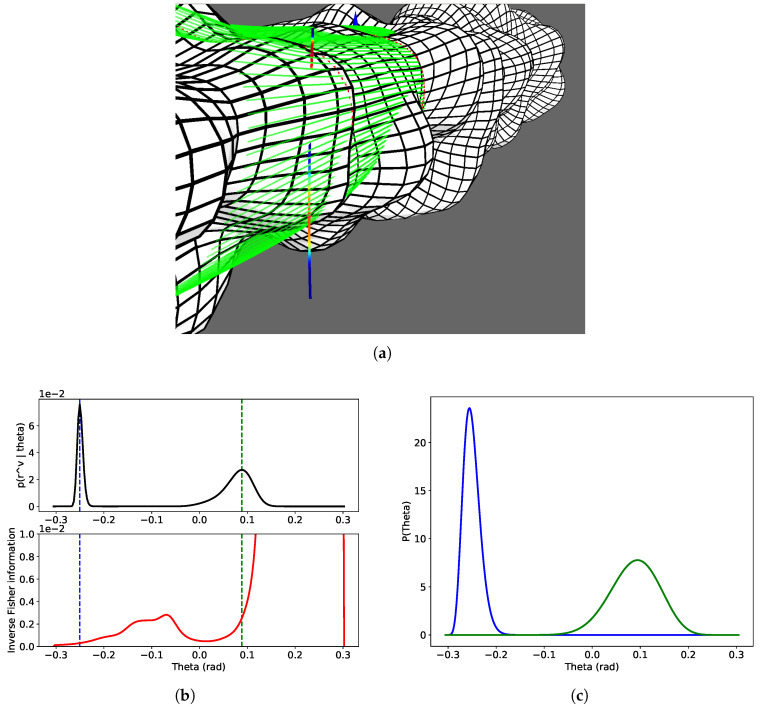
Uncertainty estimation. (**a**) Corresponding horizontal arc measurement with colors related to the beta pdf values. The top (resp. bottom) distribution correspond to the blue (resp. green) curve in the figure below. (**b**) Graphics for θ likelihood (top) and inverse fisher information (bottom). (**c**) Beta distributions corresponding to the two local maxima.

**Figure 11 sensors-20-04028-f011:**
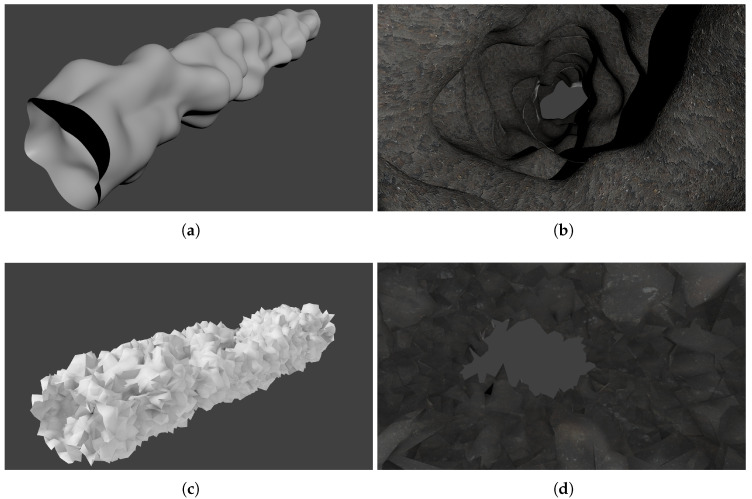
Models used in our simulations. Note that for generating sparse measurements, we used a scaled down version of the smooth karst model. (**a**) Outside view of the smooth karst model. (**b**) Inside view of the smooth karst model. (**c**) Outside view of the chaotic karst model. (**d**) Inside view of the chaotic karst model.

**Figure 12 sensors-20-04028-f012:**
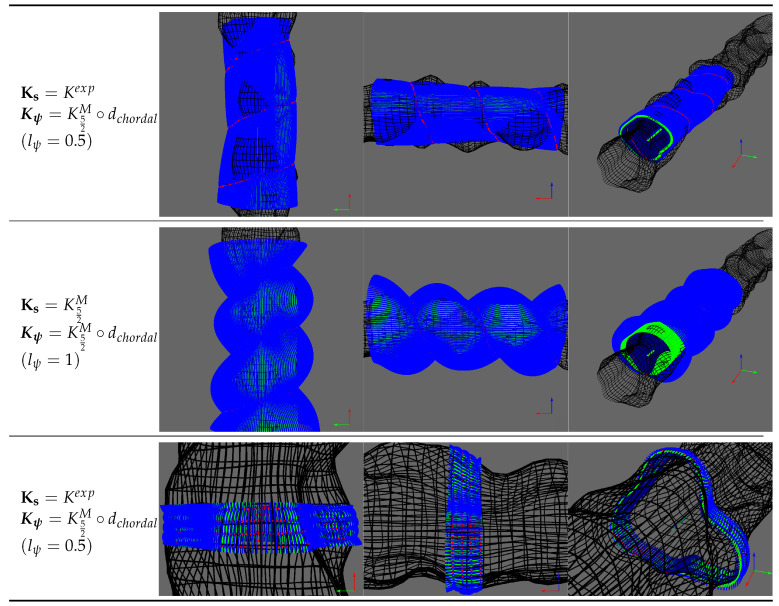
Some 3D views of the resulting GP regressions. The first two lines correspond to the sparse case and the last one to the dense case. The mean surface is represented in green and the lower/upper bound at +/− 3σ is in dark blue/blue. Red points correspond to the simulated vertical sonar measurements.

**Figure 13 sensors-20-04028-f013:**
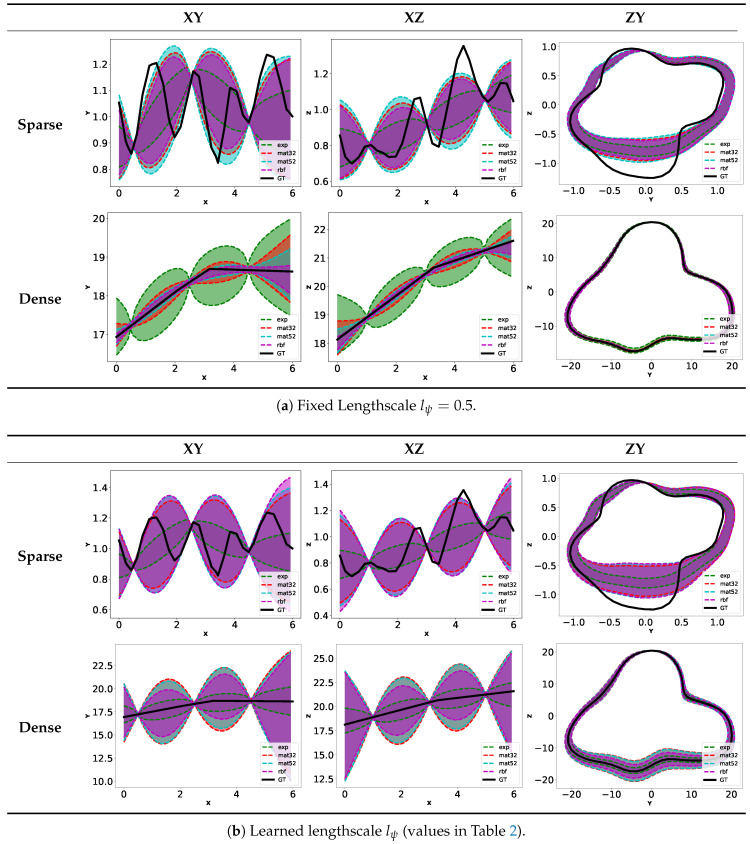
Contours of estimated surface with $K_{\psi }=K_{\frac{5}{2}}^{M}\circ d_{chordal}$ and several $K_{s}$ with different lengthscale $l_{\psi }$.

**Figure 14 sensors-20-04028-f014:**
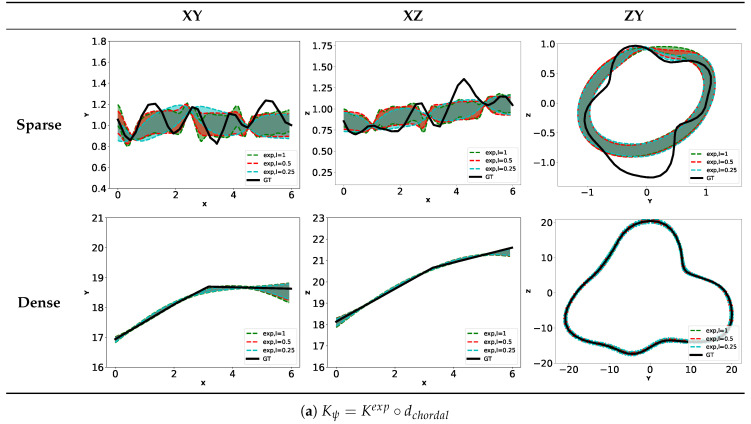
Contours of estimated surface with fixed $K_{s}=K_{\frac{5}{2}}^{M}$.

**Figure 15 sensors-20-04028-f015:**
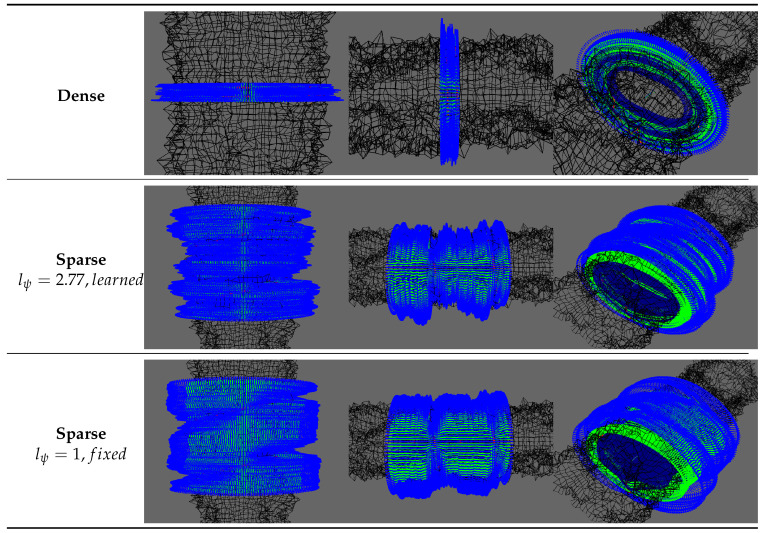
Surface estimation for a chaotic environment.

**Figure 16 sensors-20-04028-f016:**
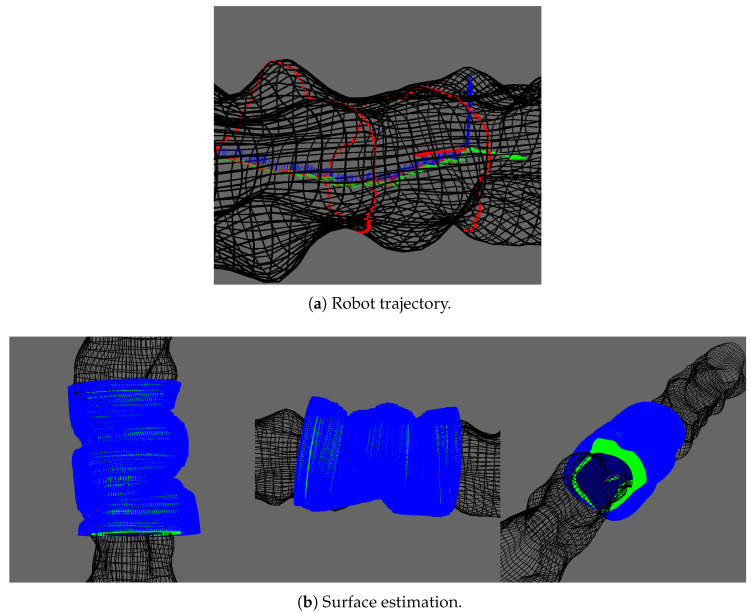
Surface estimation with a random trajectory.

**Figure 17 sensors-20-04028-f017:**
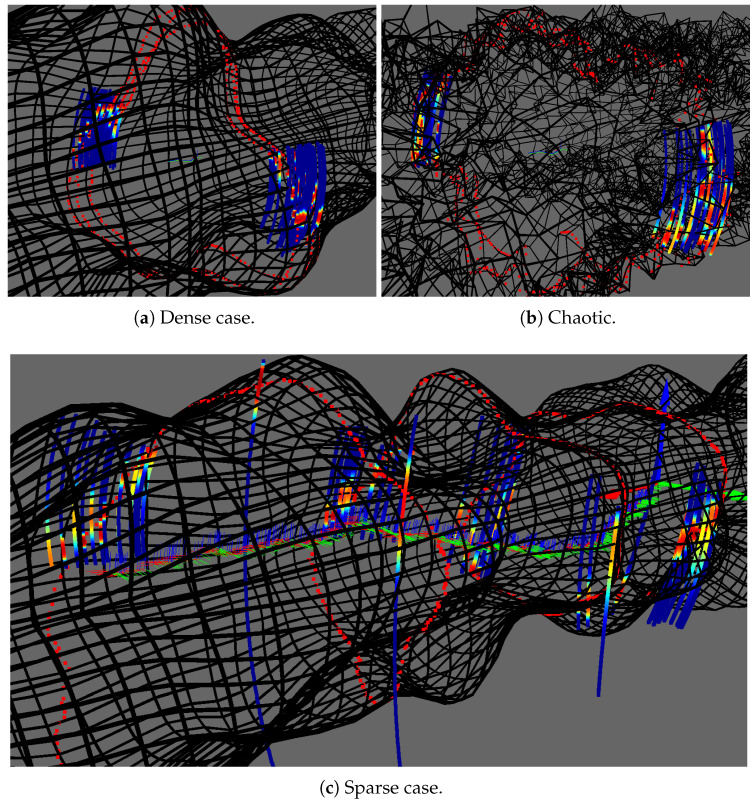
Distributions obtained in the experiment with $K_{s}=K_{\frac{5}{2}}^{M}$. For better visibility, uniform distribution are not displayed.

**Figure 18 sensors-20-04028-f018:**
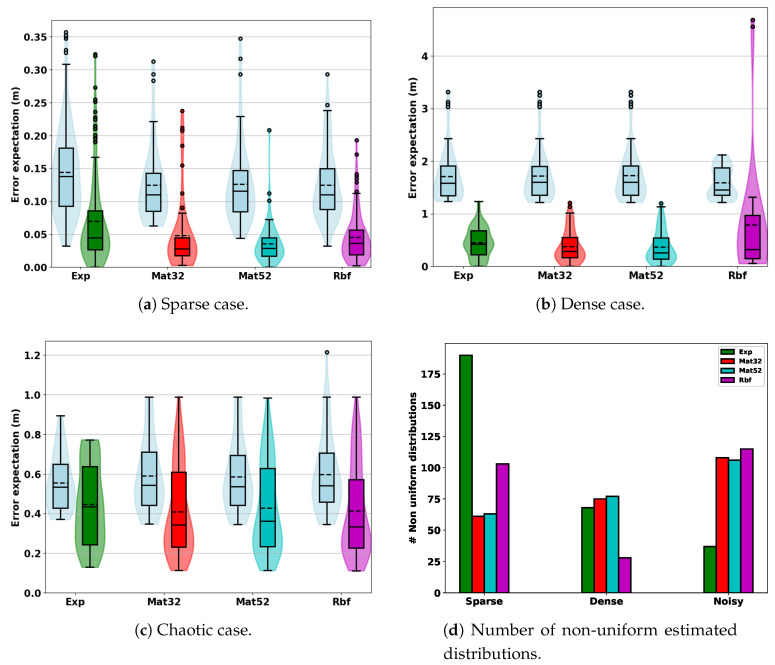
Diagrams of error distributions. The box-plot center box represents the interquartile range, IQR = Q1−Q3. The filled (resp. dot) line corresponds to the median (resp. mean). The upper (resp. lower) whisker corresponds to the largest (resp. lowest) value at a distance below (resp. above) 1.5 times the IQR. Values above and below the whiskers are outliers drawn as circles.

**Table 1 sensors-20-04028-t001:** Different settings used in our experiments.

Variable	Values
Environment	Smooth, Chaotic
Measure density	Sparse, Dense
$K_{s}$	$K^{exp},K_{\frac{3}{2}}^{M},K_{\frac{5}{2}}^{M},K^{RBF}$
$K_{\psi }$	$K^{exp}\circ d_{chordal},K_{\frac{5}{2}}^{M}\circ d_{chordal}$

**Table 2 sensors-20-04028-t002:** Lengthscale $l_{\psi }$ values in different configurations.

(**a**) Values for the two $K_{\psi }$ with $K_{s}$ fixed.		
$K_{\psi }$	**Dense**	**Sparse**
$K^{exp}\circ d_{chordal}$	9.621	75.815
$K_{\frac{5}{2}}^{M}\circ d_{chordal}$	1.029	0.699
(**b**) Values for $K_{\psi }=K_{\frac{5}{2}}^{M}\circ d_{chordal}$.		
$K_{s}$	**Dense**	**Sparse**
Exponential	0.614	0.491
Matern32	1.005	0.701
Matern52	1.016	0.709
RBF	1.018	0.770

**Table 3 sensors-20-04028-t003:** Medians/means of error distributions. Bold values corresponds to minima.

		Exp	Mat32	Mat52	RBF
**Sparse**	Prior	0.138/0.144	0.110/0.124	0.115/0.126	0.110/0.125
	Posterior	0.045/0.070	0.028/0.048	**0.028/0.035**	0.037/0.046
**Dense**	Prior	1.579/1.710	1.597/1.718	1.600/1.727	1.454/1.591
	Posterior	0.422/0.452	0.283/0.378	**0.262/0.368**	0.323/0.787
**Chaotic**	Prior	0.534/0.554	0.543/0.591	0.536/0.585	0.541/0.597
	Posterior	0.435/0.447	0.342/0.409	0.361/0.427	**0.334/0.414**

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
