# Peer review of "Elevation Angle Estimations of Wide-Beam Acoustic Sonar Measurements for Autonomous Underwater Karst Exploration"

_sensors, 2020, doi:10.3390/s20144028_

Round 1

Reviewer 1 Report

This is an interesting research paper. There are some suggestions for revision.

1. Please use one sentence to briefly discuss the existing issues of sonar measurements in abstract.

2. Please highlight your contributions in introduction.

3. In section 2, please compare pros and cons of existing solutions.

4. Between line 192 and 195, it mentions v=1/2 and v=3/2. Please explain where 1/2 and 3/2 come from.

5. In Eq. 7, what is R+?

6. In Eq. 17, please explain why tau is equal to or greater than 4.

7. In line 356, please explain why alpha = beta = 1.

8. In Eq. 45, please explain why k = 3 and how to get the proper k value.

9. In section 5, please add the experiment environment.

10. Section 5.2 is not clear. Please discuss how to obtain the optimal configurations in general.

Reviewer 2 Report

The paper describes the estimations of acoustic sonar measurements for autonomous underwater karst exploration. The problem and theoretical foundations were described, the algorithms for scanning and estimations were illustrated and the simulations were performed to validate the proposed methods.

While this paper presents fine the initial results, further work must be conducted to perform real world experiments. This can be part of a future research.

In figure 16, add also the explanations for the whiskers and the dots to the figure caption.

The conclusions section must be slightly improved. Currently it is more a summary rather than conclusions. Authors must add more conclusions and important quantitative results. For example: "We show in our experiments with different configurations that our method improves significantly the uniformly distributed prior estimation" - by how much your method  improves the results?

This paper can be accepted after the implementation of the suggestions.
